# Proteomic Analysis Revealed Different Molecular Mechanisms of Response to PEG Stress in Drought-Sensitive and Drought-Resistant Sorghums

**DOI:** 10.3390/ijms232113297

**Published:** 2022-10-31

**Authors:** Yanni Li, Binglan Tan, Daoping Wang, Yongying Mu, Guiying Li, Zhiguo Zhang, Yinghong Pan, Li Zhu

**Affiliations:** 1Biotechnology Research Institute, Chinese Academy of Agricultural Sciences, Beijing 100081, China; 2Institute of Crop Sciences, Chinese Academy of Agricultural Sciences, Beijing 100081, China; 3National Engineering Laboratory for Crop Molecular Breeding, Institute of Crop Sciences, Chinese Academy of Agricultural Sciences, Beijing 100081, China

**Keywords:** comparative proteome analysis, drought stress, differentially abundant proteins, leaf system, sorghum (*Sorghum bicolor* (L.) Moench), seedling stage

## Abstract

Drought is the major limiting factor that directly or indirectly inhibits the growth and reduces the productivity of sorghum (*Sorghum bicolor* (L.) Moench). As the main vegetative organ of sorghum, the response mechanism of the leaf to drought stress at the proteomic level has not been clarified. In the present study, nano-scale liquid chromatography mass spectrometry (nano-LC-MS/MS) technology was used to compare the changes in the protein expression profile of the leaves of drought-sensitive (S4 and S4-1) and drought-resistant (T33 and T14) sorghum varieties at the seedling stage under 25% PEG-6000 treatment for 24 h. A total of 3927 proteins were accurately quantitated and 46, 36, 35, and 102 differentially abundant proteins (DAPs) were obtained in the S4, S4-1, T14, and T33 varieties, respectively. Four proteins were randomly selected for parallel reaction monitoring (PRM) assays, and the results verified the reliability of the mass spectrometry (MS) results. The response mechanism of the drought-sensitive sorghum leaves to drought was attributed to the upregulation of proteins involved in the tyrosine metabolism pathway with defense functions. Drought-resistant sorghum leaves respond to drought by promoting the TCA cycle, enhancing sphingolipid biosynthesis, interfering with triterpenoid metabolite synthesis, and influencing aminoacyl-tRNA biosynthesis. The 17 screened important candidate proteins related to drought stress were verified by quantitative real-time PCR (qRT-PCR), the results of which were consistent with the results of the proteomic analysis. This study lays the foundation for revealing the drought-resistance mechanism of sorghum at the protein level. These findings will help us cultivate and improve new drought-resistant sorghum varieties.

## 1. Introduction

As a kind of abiotic stress, drought is the major limiting factor that directly or indirectly reduces crop productivity and inhibits crop growth [1]. In order to control the stress response and adapt to drought stress, dynamic changes in chromatin and transcriptional variation often occur in plants. Drought stress can affect plant growth and development via the germination potential, germination index, seed vigor during the germination period, relative water content (RWC) of leaves and roots, net photosynthetic rate (Pn), above-ground material accumulation, etc. [2,3,4].

Sorghum (*Sorghum bicolor* (L.) Moench) is the fifth most important grain crop in the world and is mainly distributed in arid and semi-arid tropical regions of the world [5]. Global sorghum production in 2021 was 62.167 million tons, with Africa being the largest producing region, accounting for 40% of the world’s total, followed by the United States (https://www.usda.gov/, 1 September 2022). Sorghum has a wide range of uses, such as eating, feeding, brewing, bioenergy, and chemical materials [6]. Severe drought stress will lead to shorter seedling stage, smaller ear head, poor fruit maturity, greatly reduced yield or no harvest. Drought is one of the major constraints on sorghum production, especially in African and Asian countries [7]. Severe drought stress leads to a shorter seedling stage, a smaller ear head, poor fruit maturity, a greatly reduced yield, or no harvest. Thus, it is of great significance to study the effects of drought stress on the growth and physiology of sorghum plants and to improve the drought resistance and yield of sorghum. When plants suffer from drought stress, they regulate the expression of genes and produce new proteins through the sensing and transduction of drought signals in cells, resulting in extensive morphological, physiological, and biochemical changes. To date, research on the drought resistance of sorghum has mainly focused on the physiological and biochemical aspects, and research on the molecular level is relatively rare. The natural variation of genes confers plants with different tolerances to drought stress, and most of the changes in gene expression patterns are regulated at the transcriptional level under drought stress. For example, Dugas et al. [8] performed RNA-seq analysis on the roots and shoots of sorghum genotype BTx623 seeds under PEG-induced osmotic stress or exogenous ABA stress. A total of 28,335 genes with transcriptional expression activity were obtained, among which the differentially expressed genes were involved in the biosynthesis of phytohormone, amino acid metabolism, cell growth, pathogen resistance, and ROS detoxification. In addition, signaling protein CLAVATA3, osmoprotectant β-alanine betaine biosynthetase, and water stress-inducible protein 18 (WSI18) were identified as playing important roles in the response to drought stress [8]. With the development of mass spectrometry (MS) -based proteomics, the mRNA and protein expression levels in sorghum under drought stress can be comprehensively analyzed. Mining differential proteins regulated by post-transcriptional regulation and finding and verifying some important regulatory pathways have become research focuses [9].

Proteomics is an emerging subject and hotspot in functional genomics research in the post-genomic era. It can clarify the biological functions of proteins expressed in the genome that perform life activities [10,11]. The method of two-dimensional gel electrophoresis (2-DE) combined with MS technology has been used to study the expression of sorghum proteome under different drought conditions [12], which provides a theoretical basis for revealing the drought tolerance mechanism and the drought domestication of sorghum. At present, the proteomic studies of sorghum response to drought stress mainly includes the following aspects [13,14,15,16,17]. On the one hand, the response of sorghum to drought stress is different at different developmental stages. For example, at the seedling stage, some drought-response proteins were reported to be involved in a variety of cellular functions in sorghum root, including antioxidant and defense response, carbohydrate and energy metabolism, protein synthesis and processing, transcriptional regulation, nitrogen metabolism, and amino acid biosynthesis [15]. Proteomic studies on sorghum leaves under drought stress showed that S-adenosyl-L-methionine synthase was upregulated in both drought-tolerant sorghum-11434 and drought-sensitive sorghum-11431, indicating that an increase in methionine synthase played a role in methionine activity and in maintaining osmotic regulation metabolism under drought stress. The expression of RNA-binding protein was downregulated and the RNA synthesis was inhibited in 11431, while the expression of 40S ribosomal protein S3 was upregulated only in 11434, indicating that drought-tolerant sorghum could better maintain the stability of 40S ribosomal protein and had higher RNA transcription and protein synthesis efficiency [13]. Additionally, at the flowering stage of sorghum RTx430, drought stress induces significant changes in the abundance of proteins involved in flowering time control, starch biosynthesis, rubisco activation, abscisic acid signaling, ROS scavenging, heat-shock proteins, epicuticular wax production, and phospholipid metabolism, suggesting that these proteins may play important roles in the drought tolerance of RTx430 [16]. On the other hand, different sorghum varieties showed different responses to drought at the proteomic level. For example, drought-resistant sorghum varieties upregulate detoxification/defense-related proteins, while drought-sensitive sorghum varieties downregulate the proteins involved in metabolism to cope with water stress [14,15,16,17]. Furthermore, proteomic analyses indicate that mycorrhizal and nonmycorrhizal sorghum plants use different molecular mechanisms to deal with water deficit stress [18]. It is important to reveal the expression trends of differentially abundant proteins (DAPs), within and among sorghum varieties, and their biological functions in drought resistance. Studies investigating the response to drought stress in sorghum have mainly been focused on drought-tolerant sorghum varieties, while comparative proteomics studies on drought-resistant and drought-sensitive sorghum varieties are relatively rare.

In this study, a comparative proteomic study was conducted for the first time on the leaves of drought-sensitive sorghum (S4, S4-1) and drought-resistant sorghum (T14, T33) at the seedling stage, aiming to provide support for the comprehensive proteomic evaluation of the drought response of different varieties. These four sorghum varieties were monitored for changes in their phenotypic characteristics and physiological and biochemical indicators, including RWC, chlorophyll content, superoxide dismutase (SOD) activity, catalase (CAT) activity, malondialdehyde (MDA) content, and proline (Pro) content, under drought stress simulated by 25% PEG-6000. The changes in the proteome expression in different sorghum varieties under drought stress were studied using nano-scale liquid chromatography mass spectrometry (nano-LC-MS/MS) technology. The DAPs were mainly identified by *p*-value < 0.05 and log2 fold change (FC) > 1.5, and their interaction network relationships were expounded by STRING 11.5(ELIXIR, Wellcome Genome Campus, Hinxton, Cambridgeshire, CB10 1SD, UK) and Cystoscape_v3.9.0 software (National Institute of General Medical Sciences, Bethesda, MD, USA). The purpose of this study was to elucidate the molecular basis of drought resistance differences among different sorghum varieties, explore the key candidate genes and pathways related to drought resistance in sorghum, and lay the foundations for analyzing the mechanism of drought resistance and breeding drought-resistant varieties.

## 2. Results

### 2.1. Effect of PEG Stress on the Physiology of Four Sorghum Varieties

Sorghum seedlings at the three-leaf stage were selected and treated with 25% PEG-6000 simulated drought for 24 h, and the soil volumetric moisture content of each sample was controlled at the 20% level before treatment (Appendix A). The changes in the soil water content and leaf physiological response after PEG treatment were investigated by measuring soil volumetric water content, RWC, chlorophyll content, Pro content, and SOD, CAT, and MDA enzyme activities. After 24 h of PEG treatment, the soil volumetric moisture content in groups S4, S4-1, T14, and T33 were significantly lower than those of the control groups, indicating that the soil received effective drought treatment (Figure 1A and Appendix A). The physiological changes seen in drought-sensitive and drought-resistant sorghum leaves were different under drought stress. First, the total RWC and chlorophyll content of these four varieties displayed a reduced trend after PEG treatment. Among them, the chlorophyll content of the PEG treatment groups was significantly lower than that of the control groups (Figure 1B,C and Appendix A). Then, the CAT activity and the content of Pro and MDA were increased in all sorghum varieties after PEG treatment (Figure 1D–F, and Appendix A), in which CAT activity was significantly increased (*p* = 0.0069) in drought-sensitive sorghum (S4-1) compared with controls. SOD is a vital protective enzyme in plant cells during stress. In this study, the SOD activity in drought-resistant sorghum (T14 and T33) was significantly increased (*p* = 0.0019 and *p* = 0.0310), while SOD activity in drought-sensitive sorghum (S4 and S4-1) showed different trends of increasing/decreasing (*p* = 0.0019 and *p* = 0.0310) (Figure 1G and Appendix A).

### 2.2. Phenotypic Analysis of Four Sorghum Varieties under PEG Stress

The 24 h simulated drought treatment with PEG had no significant effect on the phenotypes but had obvious effect on their physiological indexes. Thus, the samples for physiological indexes and proteomics analysis were selected after 24 h PEG treatment. Additionally, phenotypic differences were observed between the S4, S4-1, T33, and T14 sorghum varieties after 28 days of PEG-simulated drought stress. Compared with the control (drought-sensitive sorghum (S4 and S4-1)), the growth of PEG S4 showed a partially withered phenotype, with partial or complete chlorosis of leaf tips or whole leaves turning yellow to brown (Figure 2A). PEG S4-1 showed partially withered and dwarfed phenotypes, appearing shortened and stunted (Figure 2B). Compared with the control (drought-resistant sorghum (T33 and T14)), PEG T33 showed no significant differences, while T33 (Figure 2C) and PEG T14 showed partially withered and dwarfed phenotypes (Figure 2D). In general, with the prolongation of drought treatment time, the drought phenotype of sensitive varieties was more obvious than that of resistant varieties.

### 2.3. Quality Control Analysis of MS Data

A total of 3927 proteins were accurately quantitated in the sorghum seedling leaves using Proteome Discoverer software (Appendix A). To fully understand the effects of PEG treatment on the proteomics of four different sorghum varieties, the differences in Label-free quantification (LFQ) intensity between the control and PEG-treated groups were compared. The drought tolerance of different sorghum varieties was evaluated by principal component analysis (PCA) based on the LFQ intensity data collected from the control groups (S4, S4-1, T14, and T33) and PEG-treated groups (PEG S4, PEG S4-1, PEG T14, and PEG T33). The PCA results showed that PEG treatment profoundly affected the proteome expression of the PEG-treatment group, indicating that these sorghum varieties were affected by drought to different degrees (Figure 3A). The results of the quantitative protein identification statistical analysis showed that 1865, 2098, 2269, 2479, 2185, 2183, 2212, and 2144 proteins were identified in sorghum samples of S4, S4-1, T14, T33, PEG S4, PEG S4-1, PEG T14, and PEG T33, respectively (Appendix A). Four proteins were selected for parallel reaction monitoring (PRM) analysis to verify the reliability of the MS (Appendix A). Among them, HATPase_c domain-containing protein (A0A194YLP4), uncharacterized protein (C5WVD3), and ribonuclease (C5XTA6) proteins were upregulated and 3-isopropylmalate dehydratase (C5XT62) was downregulated after PEG treatment, indicating that the protein expression levels were consistent with the results of MS.

### 2.4. Statistical Analysis of DAPs

In order to identify DAPs observed in S4, S4-1, T14, and T33 sorghum varieties under drought stress compared with the control group, the *p*-value < 0.05 and up/downregulation with ± 1.5 FC was used as the filter criterion [19]. A total of 46 (14 consensus/32 PEG S4 unique proteins), 36 (7 consensus/13 S4-1 unique/16 PEG S4-1 unique proteins), 35 (8 consensus/13 T14 unique/14 PEG T14 unique proteins), and 102 (37 consensus/38 T33 unique/27 PEG T33 unique proteins) DAPs were identified, of which 46, 20, 21, and 48 DAPs were upregulated and 0, 16, 14, and 54 DAPs were downregulated, respectively (Appendix A). The consensus DAPs of different samples were visually marked by volcano plots (Figure 3B). The expression levels of DAPs in the S4 vs. PEG S4, S4-1 vs. PEG S4-1, T14 vs. PEG T14, and T33 vs. PEG T33 groups were clustered by the bidirectional clustering method. A heat map showed that the types and quantities of proteins in different sorghum varieties under PEG stress were significantly changed when compared with the control (significantly upregulated DAPs are shown in red, and those downregulated are shown in blue). Three independent biological experiments were performed in the control group and the PEG treatment group (Figure 3C).

A Venn diagram analysis of upregulated proteins in four sorghum varieties showed that there were 43, 19, 21, and 46 unique proteins in sorghum S4, S4-1, T14, and T33 after drought stress, respectively (Figure 3D). In addition, the uncharacterized protein LOC8084108 (C5XAX2) was the common upregulated protein between S4 vs. PEG S4 and S4-1 vs. PEG S4-1. Pathogenesis-related protein 10d (Q4VQB3) and pathogenesis-related protein 1-like (*PR1*) (A0A1B6QNQ4) were the common upregulated proteins between S4 vs. PEG S4 and T33 vs. PEG T33 (Figure 3B). In the downregulated proteins, there were 11, 12, and 49 unique proteins that belonged to the sorghums S4-1, T14, and T33 after drought stress, respectively (Figure 3D). Moreover, the Rieske domain-containing protein (C5WW66) was the common downregulated protein between S4-1 vs. PEG S4-1 and T14 vs. PEG T14. Eukaryotic translation initiation factor 3 subunit C (A0A1B6QDA6), uncharacterized protein (C5XCF6 and A0A1B6QJE8), and nitrate reductase (C5YM76) were shared by S4-1 vs. PEG S4-1 and T33 vs. PEG T33 Uncharacterized protein (C5YEW9) was the common downregulated protein between T14 vs. PEG T14 and T33 vs. PEG T33.

### 2.5. Pathway Enrichment Analysis of DAPs

Compared with the control group (S4 and S4-1), there were 65 upregulated and 16 downregulated proteins in the PEG S4 and PEG S4-1 samples, of which 39 upregulated and 11 downregulated proteins were annotated by the KEGG database and distributed in 79 and 16 pathways, respectively (Appendix A). In addition, there were a total of 69 upregulated and 67 downregulated proteins in the PEG T14 and PEG T33 samples, of which 29 upregulated and 45 downregulated proteins were annotated by the KEGG database and distributed in 54 and 73 pathways, respectively (Appendix A).

Then, a KEGG signal pathway enrichment analysis was performed by using the ClueGO tool in Cytoscape software (v3.8.0) with a *p*-value < 0.05 [20]. The result showed that C5XVU9, C5YE18, and C5YIC3 were enriched in the tyrosine metabolism pathway in the drought-sensitive sorghum. Additionally, there were 14 DAPs enriched in four pathways, including the citrate cycle (TCA cycle) (C5YL64, A0A1B6QPA5, A0A1Z5R408, and A0A1Z5R4Y7), sphingolipid metabolism (A0A1B6Q4S3, C5WQW4 and C5XFN1), terpenoid backbone biosynthesis (A0A1B6Q2I8, C5XCF6, C5WLY2 and A0A1W0W3N2), and aminoacyl-tRNA biosynthesis (A0A1B6Q1Q5, A0A1Z5R1W3 and C5Y8Z8) in drought-resistant sorghum (Table 1).

### 2.6. Subcellular Locations of DAPs

The correct localization of proteins in tissues and cells is of great significance to study their biological functions. In drought-sensitive and drought-resistant sorghums, the subcellular locations of 50 and 74 DAPs, respectively, annotated by the KEGG database were predicted using WoLF PSORT Protein Subcellular Localization Prediction software [21]. The functional sites in different cellular parts of DAPs in response to drought stress were analyzed by subcellular localization, reflecting the differences in the subcellular localization distribution of DAPs in drought-sensitive and drought-resistant varieties.

The results showed that DAPs were localized to chloroplast ((14-up and three-down)/(14-up and 19-down)), cytoplasm ((14-up and four-down)/(12-up and 12-down)), extracellular ((two-up and one-down)/(one-down)), mitochondria ((five-up)/(one-down)), nucleus ((three-up and two-down)/(one-up and four-down)), peroxisome ((one-down)/(one-down)), and plasma membrane ((one-up)/(six-down)) in both (drought-sensitive)/(drought-resistant) sorghums, respectively. It is worth noting that the translation initiation factor 3 subunit C (A0A1B6QDA6) protein located in the peroxisome was downregulated in both drought-sensitive and drought-resistant sorghums. In addition, the novel plant SNARE (C5WYQ2) protein localized to the Golgi was upregulated only in drought-sensitive sorghums. Additionally, the upregulated isocitrate dehydrogenase (IDH) (A0A1Z5R408) and upregulated acid phosphatase type 7 (A0A1B6PD28)/beta-fructofuranosidase (I0B6X1) proteins were localized in the cytoskeleton and endoplasmic reticulum only in drought-resistant sorghums, respectively (Figure 3E and Appendix A). The results showed that the subcellular localization distribution of DAPs was different between drought-sensitive and drought-resistant varieties.

### 2.7. Protein–Protein Interaction (PPI) Network Analysis of DAPs

In drought-sensitive sorghums, the analysis of interaction networks of a total of 31 related DAPs, among which 24 were upregulated and 7 were downregulated, were performed by using the STRING 11.5 and Cystoscape_v3.9.0 software (Figure 4A and Appendix A). Then, the top ten protein nodes were selected by CytoHubba plugin, including T-complex protein 1 subunit beta (C5XFR5), nitrate reductase (C5YM76), dihydroorotate dehydrogenase (A0A1Z5RFU2), diphosphomevalonate decarboxylase (C5XUG8), aldehyde dehydrogenase family 7 member A1 (C5XDP9), large subunit ribosomal protein L18e (C5X5J2), NADH dehydrogenase (ubiquinone) 1 alpha subcomplex subunit 6 (C5XLB7), ubiquinone biosynthesis protein COQ9 (C5WWQ6), sarcosine oxidase/L-pipecolate oxidase (C5X4K6), and glutamyl-tRNA synthetase (C5WYW2) (Figure 4C). C5XFR5, C5YM76, and A0A1Z5RFU2 were the top three important proteins ranked by the degree method [22].

Similarly, in drought-resistant sorghums, the interaction networks of 47 related DAPs, among which 20 were upregulated and 27 were downregulated, were analyzed (Figure 4B and Appendix A). The top ten protein nodes including 4-hydroxy-3-methylbut-2-en-1-yl diphosphate reductase (C5WLY2), C5YM76, U4/U6 small nuclear ribonucleoprotein SNU13 (C5WZ02), N-acetyl-gamma-glutamyl-phosphate reductase (A0A1B6QJB9), phosphoenolpyruvate carboxykinase (PEPCK) (A0A1B6QPA5), ubiquitin-small subunit ribosomal protein S27Ae (C5XIY6), IDH (A0A1Z5R408), translation initiation factor 3 subunit C (A0A1B6QDA6), T-complex protein 1 subunit alpha (C5XUD0), and 1-deoxy-D-xylulose-5-phosphate reductoisomerase (DXR) (A0A1B6Q2I8) are shown in Figure 4D, in which C5WLY2, C5YM76, and C5WZ02 were the top three most important proteins. These DAPs may potentially be involved in the response to drought stress through their interactions with each other.

### 2.8. qRT-PCR Analysis of DAPs

qRT-PCR was used to verify the reliability of the above 17 DAPs enriched in five pathways. According to the qRT-PCR results (Figure 5), the expression levels of C5XVU9, C5YE18, and C5YIC3 protein coding genes in drought-sensitive sorghum varieties (S4 and S4-1) were upregulated under PEG treatment compared with the control group, which was consistent with the results of the proteomic analysis. In the drought-resistant sorghum varieties (T33 and T14), the transcript levels of genes encoding C5YL64, A0A1B6QPA5, A0A1Z5R408, A0A1Z5R4Y7, A0A1B6Q4S3, C5WQW4, C5XFN1, A0A1B6Q2I8, C5XCF6, C5WLY2, A0A1W0W3N2, A0A1B6Q1Q5, and C5Y8Z8 proteins were differently regulated under PEG treatment compared with the control group, which were consistent with the results of the proteomic analysis, respectively. However, the transcript levels of gene encoding A0A1Z5R1W3 protein was downregulated, which was contrary to the results of proteomic analysis. This difference may be due to the influence of post-translational modification processes and the post-transcriptional regulation of genes. In general, the mRNA expression profiles of DAPs were consistent with the results of the proteomics analysis.

## 3. Discussion

### 3.1. The Effects of Drought Stress on Physiological Indicators in Drought-Sensitive and Drought-Resistant Sorghums

Moderate drought has been reported to cause stunted growth or retarded development, resulting in reduced sorghum production [23]. In this study, it was found that short-term simulated drought treatment with PEG for 24 h had no significant effect on phenotypes such as the plant height of different sorghum varieties but had obvious effects on their physiological indexes.

Compared with the control conditions, the RWC content decreased in both drought-sensitive and drought-resistant sorghum leaves under PEG stressed conditions, and the decrease in RWC in the leaves of drought-sensitive sorghums (S4: 7.80% and S4-1: 8.31%) was more than that in drought-resistant sorghum leaves (T14: 0.66% and T33: 5.42%) (Figure 2B). Previous studies have found that the RWC of the tolerant genotype was significantly higher than that of the sensitive genotype under control and drought conditions in maize (*Zea mays* L.) [24]. It is thus speculated that the less decreased RWC of drought-resistant sorghum T14 and T33 could enable them to maintain more effective biochemical and physiological processes under drought stress.

Under drought stress, the total chlorophyll content in the leaves of the four varieties decreased, and the reduction in the chlorophyll content in the drought-resistant sorghums (T14: 9.32% and T33: 9.23%) was less than that in drought-sensitive sorghums (S4: 14.26% and S4-1: 11.95%) (Figure 2C), indicating that drought-resistant sorghum showed strong adaptability to drought stress and higher photosynthetic efficiency to maintain growth. This is consistent with the gradual decrease in chlorophyll content in *Abelmoschus esculentus* under drought stress [25].

MDA is a lipid peroxidation marker, and the increased MDA content under abiotic stresses directly reflects the damage of the plant membrane system [26]. This study found that the MDA content increased in both drought-sensitive and drought-resistant sorghums under PEG stress (Figure 2F). Compared with drought-resistant sorghum (T14: 21.35% and T33: 18.39%), the content of MDA increased more in drought-sensitive sorghum (S4: 33.66%, S4-1: 24.03%), indicating that drought-resistant sorghum could better reduce the damage to the plant membrane system under drought stress. Similarly, the MDA content was also increased in the sorghum cultivar ‘Payam’ under severe and moderate drought stress conditions [27].

Pro, as a common compatible osmolyte, exists in the cytoplasm of water-stressed plants [28]. Under osmotic stress, higher Pro content can maintain the stability of the cell membrane system and prevent the dehydration of plant cells [29]. In this study, the Pro content increased in both drought-sensitive (S4: 20.19% and S4-1: 10.78%) and drought-resistant sorghums (T14: 26.58% and T33: 111.38%) under drought stress (Figure 2E). This is consistent with the increase in Pro content in sorghum bicolor inbred line BT×623 at the seedling stage under drought stress [15]. Similar trends have also been reported in that Pro accumulation significantly increased (26%) in tolerant sorghum EI9, while Pro accumulation in sensitive sorghum Tabat was decreased (5%) by compared with El9 during the water limitation periods [30]. Compared with drought-sensitive sorghum, the increase in Pro content indicated that it played an adaptive role in the stress response of drought-resistant sorghum, which enabled the membrane system to maintain a good cell osmotic regulation ability and alleviated the degree of cell membrane peroxidation under drought conditions.

Antioxidant enzymes (CAT and SOD) play an important role in eliminating peroxides and reactive oxygen species (ROS) induced by abiotic stresses, which can protect cells from damage and inhibit plasma membrane peroxidation [31]. In this study, the CAT activity both increased in drought-sensitive sorghum (S4: 31.12% and S4-1: 215.70%) and drought-resistant sorghum (T14: 167.66% and T33: 111.03%), in which S4-1 significantly increased (*p* = 0.0069) under drought stress (Figure 2D). The CAT activities of S4, S4-1, T14 and T33 were increased to varying degrees under PEG stress, indicating that both drought-resistant and sensitive sorghum had the ability to eliminate hydrogen peroxide by increasing the activity of the peroxidase enzyme, thus protecting plants from drought-stress-induced oxidative stress. The SOD activity was significantly increased (*p* = 0.0019 and *p* = 0.0310) in drought-resistant sorghum (T14: 257.82% and T33: 558.65%) and increased/decreased in drought-sensitive sorghum (S4: 2.66% and S4-1: 31.16%) (Figure 2G). The result indicated that drought-resistant sorghum has a stronger ability to control ester oxidation and reduce membrane system damage than drought-sensitive sorghum. This is the same result reported in response to drought stress by increasing ROS accumulation in sorghum bicolor inbred line BT×623 at the seedling stage [15]. In general, the drought-sensitive and drought-resistant sorghums displayed different degrees of increased CAT and SOD activities, which could increase antioxidant enzyme activity to protect cells from oxidative damage, which can effectively eliminate free radicals under simulated drought stress.

The above studies showed that, although short-term (24 h) PEG drought stress had no significant effect on the phenotype of drought-resistant and drought-susceptible sorghum varieties, it had different effects on physiological levels such as the content of RWC, chlorophyll, Pro, and MDA and SOD and CAT activity. These results will lay the foundation for further understanding the resistance mechanism of different resistance sorghum to drought stress.

### 3.2. Effects of Drought Stress on Defense Mechanism-Related DAPs in Drought-Sensitive Sorghum

In this study, the response of drought-sensitive sorghum leaves to PEG stress was mainly enriched in the tyrosine metabolism pathway, among which the three proteins C5XVU9, C5YE18, and C5YIC3 were significantly upregulated, which played an important role in the drought stress response.

Drought stress can increase ROS free radicals in plants and cause cells to suffer from oxidative stress. In order to remove the excess ROS to maintain normal plant growth, plants have developed elaborate and complex defense systems, such as tyrosine metabolism [32]. It has been reported that the tyrosine metabolic pathway responds to biotic and abiotic stress responses by participating in the scavenging of ROS [33]. Tyrosinase is a multifunctional copper-containing oxidase, which belong to the family of polyphenol oxidases (PPOs) [34]. As an important oxidoreductase, PPO can catalyze the oxidation of moniphenols and o-diphenols and plays a crucial role in scavenging active oxygen, the biosynthesis of aurones and betaine secondary metabolites, and the enhancement of the plant’s resistance to stress [35,36]. Previous studies have shown that the activities of PPO decreased during the progression of drought stress in all tissues of olive trees [37]. However, PPO activity increased under drought, salinity, and drought + salinity stresses in pistachio rootstocks [38] and also increased in maize under drought stress [39]. Overall, decreasing or increasing changes in PPO activity occurred in response to drought stress by up- or downregulating the expression of PPO proteins. In this study, the expressions of PPOs (C5YE18 and C5YIC3) were upregulated in drought-sensitive sorghum, which may limit the damage of ROS to cells during water deficit. The result showed that the changes in PPO expression might be an important attribute linked to sorghum drought tolerance.

Furthermore, the tyrosine metabolism pathway was enriched under drought stress, in which the S-(hydroxymethyl) glutathione (HMGSH) dehydrogenase protein (C5XVU9) was significantly upregulated in drought-sensitive varieties. It has been reported that HMGSH dehydrogenase-mediated nitric Oxide (NO) metabolism plays an important role in regulating normal physiological processes and host defense in plants. For example, the enzyme is involved in redox homeostasis during magnaporthe grisea development and host infection of *M. oryzae*, and S-nitrosothiols (S-Nos) homeostasis and active nitrogen metabolism in *Solanum lycopersicum* cells [40,41]. Therefore, drought-sensitive sorghum may enhance the defense system to cope with drought stress through the tyrosine pathway, in which PPOs (C5YE18 and C5YIC3) and HMGSH dehydrogenase (C5XVU9) proteins play important roles.

### 3.3. Effects of Drought Stress on DAPs Related to TCA Cycle in Drought-Resistant Sorghum

The tricarboxylic acid (TCA) cycle is an important metabolic pathway that unifies lipid, carbohydrate, and protein metabolism. Oxaloacetic acid, as an intermediate involved in the TCA cycle, can flux out of the cycle and be catalyzed by the decarboxylation enzyme PEPCK, thus leading to a reduction in carbon flow for fatty acid biosynthesis [42]. The TCA cycle also plays a role in plant defense responses, such as IDH product of NADH by the TCA cycle to promote redox signaling linked to pathogen responses [43]. In the next step, dihydrolipoamide acetyltransferase (DLAT) in pyruvate dehydrogenase complex (PDC) effectively eliminates NADH, which has lower maximum activity and represents a bottleneck of the TCA cycle [44]. NADH exports by establishing conditions for the operation of the citrate valve and contributes to the biosynthesis of amino acids and other metabolic products during photosynthesis [45].

In this study, PEPCK(A0A1B6QPA5), IDH (A0A1Z5R408), and DLAT (A0A1Z5R4Y7 and C5YL64) were identified to be enriched in the TCA cycle pathway of drought-resistant sorghum under drought stress. PEPCK upregulation may help to exacerbate the catalytic action of oxaloacetic acid, resulting in reduced fatty acid biosynthesis. The upregulated IDH and downregulated DLAT may lead to more NADH production and less NADH consumption, which contribute to the enhancement of amino acid metabolism during photosynthesis and thus enhance resistance to drought stress. Therefore, the enhanced TCA cycle activity could provide more energy for the synthesis of amino acids and metabolites in sorghum to adapt to drought stress.

### 3.4. Drought-Stress-Affected DAPs Related to Sphingolipid Metabolism in Drought-Resistant Sorghum

Lipids, as major components of biological membranes, mediate lipid signaling in response to various abiotic stresses, including drought, pathogen attack, salinity, and cold [46]. Lipid metabolism is regulated by multiple signaling pathways and generates a variety of bioactive lipid molecules. Lipid signaling molecules encompass various lipid classes such as fatty acids, sphingolipids, lysophospholipids, phosphatidic acids, and diglycerides [47,48,49,50]. Lipids as signaling mediators are typically involved in plant defense responses through enhanced sphingolipid synthesis, and as stress relievers to reduce the intensity of stressors [47]. Sphingolipids are a class of lipids containing a backbone of sphingoid bases. Sphingolipid metabolites regulate cellular processes, including programmed cell death. The modulation of sphingolipid biosynthesis and catabolism has been reported to improve plant tolerance to both biotic and abiotic stresses [51].

In the reported differential proteomic analysis of sorghum bicolor (BTx623) root response to simulated drought stress, two proteins associated with lipid metabolism (patatin (spot 5516) and lipoxygenase (spot 7806)) enhanced their abundance under drought stress. Upregulated patatin could provide energy for roots during drought stress by degrading lipids. Lipoxygenase functions in response to lipid biosynthesis and fatty acid metabolism, which was also reported to be upregulated in the roots of barley under salt stress [15,52]. It was also found that upregulated Acyl carrier proteins were identified as stress responsive proteins in EI9-tolerant and Tabat-sensitive sorghum accessions. Acyl carrier protein has the effect of synthesis and subsequent desaturation and acyl transfer of fatty acids in plants and bacteria [30]. At present, the DAPs in response to drought have been reported to focus on fatty acid metabolism, while the drought-response proteins involved in sphingolipid metabolism have not been reported. In this study, three DAPs were found to be enriched in sphingolipid metabolic pathways in drought-resistant sorghum, including neutral ceramidase (nCDase) (A0A1B6Q4S3), serine palmitoyltransferase (SPT) (C5WQW4), and S1P aldolase (C5XFN1). It has been reported that the nCDase protein is one of the vital enzymes involved in the biosynthesis of sphingolipids, which can hydrolyze ceramide into sphingosine and fatty acids [53]. Over-expressing nCDase AtNCER1 showed enhanced tolerance to oxidative stress in *Arabidopsis*. The research indicated that nCDase can affect sphingolipid homeostasis and oxidative stress responses [54]. Under drought stress, the abundance of nCDase (A0A1B6Q4S3) was downregulated in drought-resistant sorghum, indicating that nCDase may play different roles in response to drought stress in different crops. SPT is the central control point of bioactive sphingolipid synthesis and plays an important role in mediating cellular stress response [55]. A SPT (C5WQW4) was found to be upregulated in drought-resistant sorghum, which may improve the response ability of sorghum to drought stress. S1P aldolase, also known as sphingosine-1-phosphate Lyase (SPL) [56], catalyzes the last step in sphingolipid degradation and is a key enzyme in regulating the intracellular and circulating levels of long-chain base phosphate (LCBP). Sphingolipid LCBP plays an important role in cell interaction, cell proliferation, and cellular stress response, and it has been reported to be involved in signal transduction during drought [57,58]. In this study, the expression levels of nCDase (A0A1B6Q4S3), SPT (C5WQW4), and S1P aldolase (C5XFN1) proteins were significantly altered, suggesting that these proteins may mediate the cellular stress response by affecting sphingolipid homeostasis, thus improving the tolerance of sorghum to oxidative stress.

### 3.5. Drought-Stress-Affected DAPs Related to Terpenoid Backbone Biosynthesis in Drought-Resistant Sorghum

Plant terpenoids are the class of natural products with the most structural changes in plants. Their synthesis in organisms can be caused by isoprene end-to-end formation or by isoprene ring formation [59]. Isoprene first needs to be activated and converted into isopentenyl pyrophosphate (IPP) and dimethylallylpyrophosphate (DMAPP). IPP and DMAPP can be generated through the mevalonate (MVA) and methylerythritol-4-phosphate (MEP) pathway. The MVA pathway is mainly used for the synthesis of sesquiterpenes, triterpenes, sterols, brassinosteroids, polyterpenes, and the moieties used for prenylated proteins in plant cytoplasm. The MEP pathway mainly provides precursors for monoterpenes, diterpenes, and tetraterpenes in plant plastids (such as carotenoids, abscisic acid (ABA), the phytohormones gibberellins phytol, tocopherols, the side chain of chlorophylls, phylloquinones, and plastoquinone) [60,61]. Terpenoid backbone biosynthesis is mainly responsible for the synthesis of different terpenoids. Terpenoids affect multiple aspects of plant growth, development, and stress response by modulating phytohormone metabolism and phytosterol content [62].

In this study, a total of four DAPs enriched in terpenoid backbone biosynthesis were firstly screened from drought-resistant sorghum under drought stress, including DXR (A0A1B6Q2I8), GGPS (C5XCF6), HDR (C5WLY2), and DHDDS (A0A1W0W3N2). Among them, DXR is a key enzyme in triterpenoid metabolic synthesis, which catalyzes the second step of the MVA pathway to form MEP [63]. The GGPS catalyzes the synthesis of the 20-carbon isoprenoid geranylgeranyl diphosphate (GGPP), which is the precursor substance of the diterpene synthesis pathway [64]. Additionally, HDR is the last key enzyme in the MEP pathway to synthesize isopentenyl diphosphate [65]. DHDDS catalyzes the elongation of the cis-prenyl chain to produce the polyprenyl backbone of dolichol, which is a class of essential polyisoprenoid within the endoplasmic reticulum of all eukaryotes and plays crucial roles in protein glycosylation [66,67]. In this study, DXR, GGPS and HDR proteins were all downregulated in drought-resistant sorghum varieties, indicating that the MVA and MEP metabolic pathways in sorghum may have a negative response to drought stress. In addition, the expression of DHDDS was upregulated under drought stress in drought-resistant sorghum varieties, which may promote the production of the polyprenyl backbone of dolichol. Dolichol affected protein transport in the endoplasmic reticulum and had a potential resistance to endoplasmic reticulum stress [68]. The results showed that drought stress affected the abundance of DXR, GGPS, HDR, and DHDDS related to terpenoid backbone biosynthesis in response to drought stress in drought-resistant sorghum.

### 3.6. Drought-Stress-Affected DAPs Related to Aminoacyl-tRNA Biosynthesis in Drought-Resistant Sorghum

Plants can respond to abiotic stress by inhibiting protein biosynthesis and increasing levels of molecular chaperones that control protein folding and processing. In protein biosynthesis, the aminoacyl-tRNA synthetases (aaRSs) are essential enzyme-linking amino acids to tRNAs, which provide the building blocks for ribosomal protein synthesis [69]. It is well documented that aaRSs participate in cellular stress responses in bacteria. Aspartyl-tRNA synthetase (AspRS) is a kind of aaRSs mediating the perception of β-aminobutyric acid (BABA). As a non-proteinaceous amino acid, BABA can protect plants from various abiotic stresses and provide broad-spectrum disease protection. BABA interferes with AspRS canonical activity resulting in the activation of cellular defense mechanisms [70]. Histidyl-tRNA synthetase (HARS) is also a member of the aaRSs family, which is responsible for the synthesis of histidine transfer RNA and plays an important role in the binding process of histidine to protein [71,72]. In this study, AspRS (A0A1B6Q1Q5 (downregulated) and C5Y8Z8 (upregulated)) and HARS (A0A1Z5R1W3) (downregulated) were enriched in aminoacyl-tRNA biosynthesis in drought-resistant sorghum under drought stress. The expression trends of A0A1B6Q1Q5 and C5Y8Z8 protein in response to drought were the opposite of one another, which may be due to the different expression characteristics of proteins with different amino acid compositions. It has been reported that the aspartate-tRNA protein is also involved in protein synthesis in the roots of sorghum BT×623 response to PEG-induced drought stress at the seedling stage, in which aspartate–tRNA ligase 2 cytoplasmic (spot 6704) was downregulated [15]. However, HARS proteins have not been reported in sorghum. These results suggest that drought stress could reduce the abundance of AspRS and HARS and repress protein synthesis, processing, and turnover in sorghum.

### 3.7. Comparative Analysis of Drought Stress Responses between Drought-Sensitive and Drought-Resistant Sorghums

This research mainly investigated the resistance mechanism of drought-sensitive and -resistant sorghum leaves under 24 h PEG-simulated drought stress using proteomic analysis technology. The results showed that 81 and 136 DAPs were identified in drought-sensitive and drought-resistant sorghum, respectively. Among the DAPs, it was found that following drought stress treatment, pathogenesis-related protein 10d (PR-10) (Q4VQB3) and pathogenesis-related protein 1-like (PR1) (A0A1B6QNQ4) were upregulated in both drought-sensitive and drought-resistant sorghum varieties (Figure 3D). PR-10 is a pathogenic protein [73] that has been implicated in the defense response of sorghum. It has been found that PR-10 played a positive role in preventing fungal colonization and grain mold, and its expression in glumes of resistant sorghum varieties (Tx2911 and Sureno) was more strongly induced than in that of susceptible sorghum varieties (RTx430 and SC170-6-17) [74]. In addition, the rapid induction of root-specific rice PR10 (RSOsPR-10) under salt- and drought-stress may possibly be achieved by activating the jasmonic acid signaling pathway. These results indicate that RSOsPR-10 can protect rice from salinity and drought stress, which is of great significance in the genetic engineering of plants’ response to water-deficiency stress [75]. PR1 is also considered to be an important defense protein with antifungal activity [76]. The SlPR-1 gene has been reported to be upregulated under drought stress in both *Fusarium oxysporum* tolerant and sensitive tomato varieties, supporting the idea that it plays a positive role in drought response [77]. However, the function of pathogenesis-related protein in sorghum response to drought stress has not been reported in sorghum. In this study, PR-10 (Q4VQB3) and PR1 (A0A1B6QNQ4) were activated in both drought-sensitive and drought-resistant sorghum varieties under drought stress, indicating that they may have important functions in the drought resistance of sorghum and that they require further study.

Moreover, Rieske domain-containing protein (C5WW66), eukaryotic translation initiation factor 3 (*eIF3*) (A0A1B6QDA6), GGPS (C5XCF6), nitrate reductase (C5YM76), and subtilisin-like protease SBT1.7 (A0A1B6QJE8) proteins were all downregulated in both drought-sensitive and drought-resistant sorghum varieties under PEG-simulated drought stress (Figure 3D). It has been reported that the Rieske domain-containing protein (Rieske 2Fe-2S) plays an essential role in electron transfer and transmembrane charge transfer, which catalyzes the oxidoreduction of ubiquinol and cytochrome [78]. However, no relevant studies on this protein in sorghum have been reported. In this study, the protein C5WW66 was found to be downregulated in both drought-sensitive and drought-resistant sorghums under the influence of drought stress. *eIF3* has been reported to play a role in abiotic stress resistance and also has not been researched in sorghum. In *Arabidopsis*, the overexpressed *eIF3* plants exhibited a significantly higher survival rate, soluble protein content, and photosynthetic efficiency, and enhanced the ability to protect against photooxidative stress under drought conditions [79]. Interestingly, *elF3* was downregulated in drought-sensitive and drought-resistant sorghum varieties under drought stress. This suggests that this protein may play different roles in the drought resistance mechanism of sorghum, and its function needs to be further studied. GGPS (C5XCF6) was downregulated in drought-sensitive and drought-resistant sorghum varieties, indicating that the MVA metabolic pathways of sorghum may show a negative response to drought stress. Nitrate reductase is an enzyme involved in N metabolism. Under drought stress, the photosynthesis, total nitrogen accumulation, and root growth were weakened in *Malus prunifolia*, and the activity of nitrate reductase was significantly reduced [80]. In this study, nitrate reductase (C5YM76) was found to be downregulated in both drought-resistant and drought-sensitive sorghum, indicating that drought stress had effects on the photosynthesis and nitrogen source metabolism of drought-resistant and drought-sensitive varieties, especially on the decrease in chlorophyll content (Figure 1C). Subtilisin-like protease (SBT) SBT1.7 is a specific family of serine peptidases, in which SBT1.4, SBT3.7, and SBT3.8 were found to be upregulated in response to osmotic stress [81]. However, under drought stress, the downregulation of SBT1.7 (A0A1B6QJE8) may have an effect on the osmoregulation of drought-sensitive and drought-resistant sorghum varieties.

These seven proteins might reflect the commonality of metabolic changes in resistance to drought stress, suggesting that they may participate in or regulate the different responses of drought-resistant and drought-sensitive varieties to drought stress, and their functions need to be further studied.

Then, the KEGG signal pathway enrichment analysis was performed on 81 and 136 DAPs identified in drought-sensitive and drought-resistant sorghum, respectively. Among the enriched pathways, three DAPs including PPOs (C5YE18 and C5YIC3) and HMGSH dehydrogenase protein (C5XVU9) in a drought-sensitive varieties were mainly enriched in tyrosine metabolism pathway, which enhanced the defense system to cope with drought stress. In addition, 14 DAPs in drought-resistant varieties were mainly enriched in four pathways, which responded to drought stress by altering the abundance of proteins involved in the processes related to amino acid, energy, defense/detoxification, sphingolipid, and terpenoid biosynthesis. Firstly, PEPCK (A0A1B6QPA5), IDH (A0A1Z5R408) and DLAT (A0A1Z5R4Y7 and C5YL64) were mainly enriched in the TCA cycle pathway, indicating that enhanced TCA cycle activity could provide energy for various amino acid metabolism and photosynthesis in sorghum to tolerate drought stress. Then, nCDase (A0A1B6Q4S3), SPT (C5WQW4), and S1P aldolase (C5XFN1) were enriched in sphingolipid metabolism, indicating that drought stress may affect sphingolipid homeostasis and oxidative stress response and then affect the tolerance of sorghum to drought stress. Next, AspRS (A0A1B6Q1Q5 and C5Y8Z8) and HARS (A0A1Z5R1W3) were enriched in Aminoacyl-tRNA biosynthesis, indicating that drought-resistant sorghum possibility responded to drought stress by affecting amino acid metabolism and protein synthesis. Finally, DXR (A0A1B6Q2I8), GGPS (C5XCF6), HDR (C5WLY2), and DHDDS (A0A1W0W3N2) were enriched in terpenoid backbone biosynthesis, indicating that the MVA and MEP metabolic pathways in sorghum may have a negative response to drought stress and have a potential resistance to endoplasmic reticulum stress.

In this study, the proteomic analysis of drought-resistant (T33 and T14) and drought-sensitive sorghum (S4 and S4-1) sorghum varieties leaves at the seedling stage in response to PEG-simulated drought stress is summarized in Figure 6. A total of 81 DAPs (65 upregulated and 16 downregulated) were identified in drought-sensitive sorghum varieties, of which three were enriched in the tyrosine metabolism pathway with defense functions that play an important role in drought-sensitive sorghum varieties’ response to drought stress. Additionally, 136 DAPs (69 upregulated and 67 downregulated) were identified in drought-resistant sorghum varieties, in which 14 DAPs enriched in citrate cycle (TCA cycle), sphingolipid metabolism, terpenoid backbone biosynthesis, and aminoacyl-tRNA biosynthesis pathways, respectively, may contribute to drought-resistant sorghum varieties to copy drought stress by altering protein abundances involved in processes related to amino acids, energy, defense/detoxification, sphingolipids, and terpenoid biosynthesis.

## 4. Materials and Methods

### 4.1. Plant Materials and Growth Conditions

The drought-sensitive (S4 (Heibeinuoza 5) and S4-1(Qianjinchui)) and drought-resistant (T33 (Theis) and T14 (Rio)) sorghum varieties were obtained from the Germplasm Bank of the Institute of Crop Sciences, Chinese Academy of Agricultural Sciences. Additionally, their ability to respond to drought stress has been identified by multilevel phenotypic analysis. The screening indicators and identification methods of drought resistance of sorghums were described in our previous work [82]. The seeds of each sample were soaked in distilled water for 24 h in the dark at 25 °C, and then the 30 germinating seeds were transplanted into pots (pot size: 19 × 14.2 × 11cm in length, width, and height, respectively, 1000 g nursery media, sowing depth 1 cm) with 150 mL of distilled water. Seedlings were cultured in a controlled plant growth chamber (16 h light/8 h dark, 28 °C, 60% relative humidity). We added 50 mL of distilled water every three days. At the three-leaf stage, the soil volumetric moisture content of all samples was determined to be consistent (around 20%) using an AZS-100 Soil Moisture Handheld Meter (Beijing Aozuo Ecology Instrumentation Ltd., Beijing, China) before treatment. Then, the experimental groups were treated with 150 mL 25% PEG-6000 for 24 h, and the control groups were treated with 150 mL distilled water for 24 h. The leaves of each group were sampled and frozen at −80 °C for the subsequent determination of physiological indexes and proteomics analysis. The seedlings of the control group and PEG-treated group continued to be cultured with 50 mL distilled water and 25% PEG-6000 every three days, respectively. The phenotypic changes of plants were recorded on the 28th day.

### 4.2. Determination of Physiological Indicators

The leaf RWC was determined according to Galmés et al. [83]. The chlorophyll content was measured using a SPAD 502 meter (Konica-Minolta, Japan). The SOD activity was determined by a SOD activity detection kit (Solarbio, Beijing, China). The CAT activity was determined by a CAT activity assay kit (Solarbio, Beijing, China). The MDA content was determined using an MDA content detection kit (Solarbio, Beijing, China), and the content of Pro was quantified according to the instructions of a Pro content detection kit (Solarbio, Beijing, China). Compared with the control, the increase/decrease ratio of physiological indexes including the content of RWC, chlorophyll, and Pro, and the enzyme activity of SOD, CAT, and MDA were calculated using the formula: (experimental group − control group)/control group × 100% (n = 3).

### 4.3. Protein Extraction and Digestion

Sorghum leaves of S4, S4-1, T14, T33, PEG S4, PEG S4-1, PEG T14, and PEG T33 (200 mg/sample, three biological repeats) were used for protein extraction. The leaves were powdered using a Cell Disruption System (Retsch, Haan, Germany), and the extraction method of protein was performed according to Zhu et al. [84]. Then, the extracted proteins were digested by 50 μL trypsin (Promega, Madison, WI, USA), incubated overnight at 37 °C, centrifuged at 13,000× *g* for 30 min, and stored at −20 °C for subsequent analysis.

### 4.4. Nano Mass Spectrometric Analysis

Tryptic peptides were identified by nano-liquid chromatography (nLC) and tandem MS using an EASY-nLC 1000 system coupled to a Q Exactive Plus Orbitrap mass spectrometer (Thermo Fisher Scientific, San Jose, CA, USA). The parameters of the mass spectrometer were set according to Zhu et al. [84]. The experiments were repeated at least three times for each sample, and we obtained the raw MS files with high reliability (Appendix A).

### 4.5. Protein Identification and Label-Free Quantification

The LFQ analysis of proteins was performed using MaxQuant software (version 1.6.0, http://www.coxdocs.org, 1 June 2019) and the Uniprot database (https://www.uniprot.org/, 1 June 2019). The parameters were set as follows: high confidence of peptide fragmentation; 20 ppm precursor mass tolerance; 0.02 Da fragment ion tolerance; up to two missed cleavages; carbamidomethyl cysteine as a fixed modification; N-terminal acetylation and PPDK oxidation of proteins were defined as variable modifications. Three biological and technical replicates were performed for protein identification and LFQ. The quantitative results obtained from the raw files were used for the subsequent bioinformatics analysis.

### 4.6. Bioinformatics Analysis

The qualitative and LFQ data were pre-processed with a standardized protocol before being used for screening related proteins. R package (ggord) was used to calculate the PCA for four sorghum varieties under normal and PEG-treated conditions, to perform a cluster analysis, and to plot the LFQ intensity dataset. The DAPs were considered statistically significant at *p*-value < 0.05 and |log2 FC| ≥ 1.5 compared to the drought stress treatment with the control treatment [19]. Consensus DAPs of S4/PEG S4, S4-1/PEG S4-1, T14/PEG T14, and T33/PEG T33 were marked by volcano plots with R package (ggplot2 and ggthemes). The DAPs were illustrated by heat maps in R package (pheatmap and ggplot2). Venny 2.1 (https://bioinfogp.cnb.csic.es/tools/venny/, 5 September 2022) was used to describe the DAPs between different varieties and different drought treatments. The functional annotation and classification of the identified DAPs were performed by KEGG BlastKOALA (https://www.kegg.jp/blastkoala/, 9 September 2022). Then, the KEGG pathway enrichment analysis was performed using the ClueGO tool in the Cytoscape software (v3.8.0) [20,85]. The subcellular location prediction of DAPs was conducted followed by WoLF PSORT Protein Subcellular Localization Prediction (https://wolfpsort.hgc.jp/, 13 September 2022). The protein–protein interactions of the DAPs were performed to determine their functions and pathways using STRING 11.5 (https://cn.string-db.org/, 17 September 2022) and align species as *Sorghum bicolor* (L.) Moench with default parameters. The CytoHubba plugin of the Cytoscape software was used to sort and find the top ten key protein nodes [22].

### 4.7. PRM Analysis

The PRM analysis of certain DAPs was performed on a Q-Exactive Plus mass spectrometer equipped with an Easy nLC-1000 system (Thermo Fisher Scientific, Waltham, MA, USA). The parameters of MS were set according to Wang et al. [86].

### 4.8. RNA Extraction and qRT-PCR Analysis

The total RNA was extracted from each sample using TRNzol Universal Reagent (TIANGEN, Beijing, China) according to the manufacturer’s instructions. cDNA was synthesized by All-In-One 5X RT Master Mix following the manufacturer’s protocol (ABM, Vancouver, BC, Canada). qRT-PCR was performed on the Applied Biosystems 7500 (Applied Biosystems, Foster City, CA, USA) using the Taq Pro Universal SYBR™ qPCR Master Mix (Vazyme, Beijing, China). The gene-specific primers used for qRT-PCR were designed using the Primer Blast tool in NCBI according to the cDNA sequences obtained from the Sorghum Genomics Database (v3.1.1). The sorghum actin gene was used as an endogenous control for normalization. Briefly, RT was carried out using 100 ng total RNA. The PCR reaction was carried out in 20 μL volume, containing 10 μL Mix reagent, 0.5 μM each forward and reverse primer, 2 µL diluted cDNA and 7 μL sterile distilled water, with the following protocol: denaturation at 95 °C for 30 s; followed by 42 cycles of 95 °C for 10 s; annealing at 60 °C for 15 s; extension at 72 °C for 25 s. All reactions were carried out on three technical replicates for each biological replicate. Data analysis was carried out using the Applied Biosystems 7500 version software (ABI) and the relative gene expression was calculated using the 2^−ΔΔCt^ method [87]. The primers of all genes are listed in Appendix A.

A flowchart of the proteomics analysis of drought-sensitive (S4 and S4-1) and drought-resistant (T33 and T14) sorghum varieties is illustrated in Appendix A.

## 5. Conclusions

In this study, drought-resistant sorghums (T14 and T33) and drought-sensitive sorghums (S4 and S4-1) were selected to study the differences in the phenotype, physiology, and proteomics of different sorghum variety leaves at the seedling stage by PEG-simulated drought stress. After 24 h of PEG treatment, the soil volumetric moisture, RWC, and chlorophyll content of all four samples were decreased, and the Pro and MDA content and CAT activity were increased when compared with the control, although there was no significant difference in the plants’ phenotype, such as plant height. Nano LC-MS/MS technology was used to identify the proteome expression of drought-sensitive and drought-resistant sorghums under drought stress. Compared to controls, the response mechanism of drought-sensitive sorghum to drought was attributed to the upregulated antioxidant enzyme PPO can eliminate ROS production and alleviate oxidative stress; the upregulated S-(hydroxymethyl)glutathione dehydrogenase can enhance the defense capability of sorghum by expressing nitric oxide (NO)-mediated metabolism, while compared with the controls, the response mechanism of drought-resistant sorghum to drought can be revealed: promoting the TCA cycle, enhancing sphingolipid biosynthesis, interfering with triterpenoid metabolite synthesis by differentially expressing the proteins DXR, GGPS, HDR, and DHDDS, and influencing aminoacyl-tRNA biosynthesis by differential expression of aspartyl-tRNA/histidyl-tRNA synthetase. Through a series of proteomic analyses, a total of 17 important candidate proteins related to drought stress were screened and verified by qRT-PCR. Further studies will focus on verifying the specific biological functions and mechanisms of these 17 key candidate proteins in response to drought stress in sorghum so as to provide genetic resources for the molecular breeding of sorghum drought resistance. In the future, proteomics can be combined with post-translational modification omics, transcriptomics, and metabolomics to deeply explore species characteristics and signal regulatory networks.

## Figures and Tables

**Figure 1 ijms-23-13297-f001:**
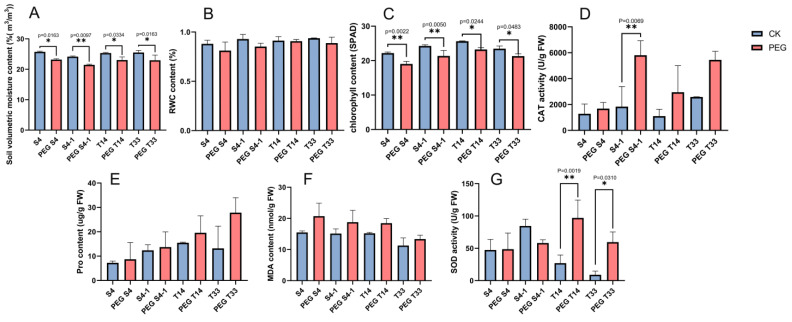
Physiological characteristics of drought-sensitive and drought-resistant sorghums under control and PEG stressed conditions. Measurements of the soil volumetric moisture (%(m^3^/m^3^)) (**A**), RWC (%) (**B**), chlorophyll (SPAD) (**C**), CAT activity (U/g FW) (**D**), Pro (ug/g FW) (**E**), MDA (nmol/g FW) (**F**), and SOD activity (U/g FW) (**G**). Bars represent the mean ± SE (n = 3). Data represent the mean ± variance of three biological replicates, and statistical significance was measured using one-way ANOVA with Tukey’s test. * *p* < 0.05 and ** *p* < 0.01 between treatments and their respective controls.

**Figure 2 ijms-23-13297-f002:**
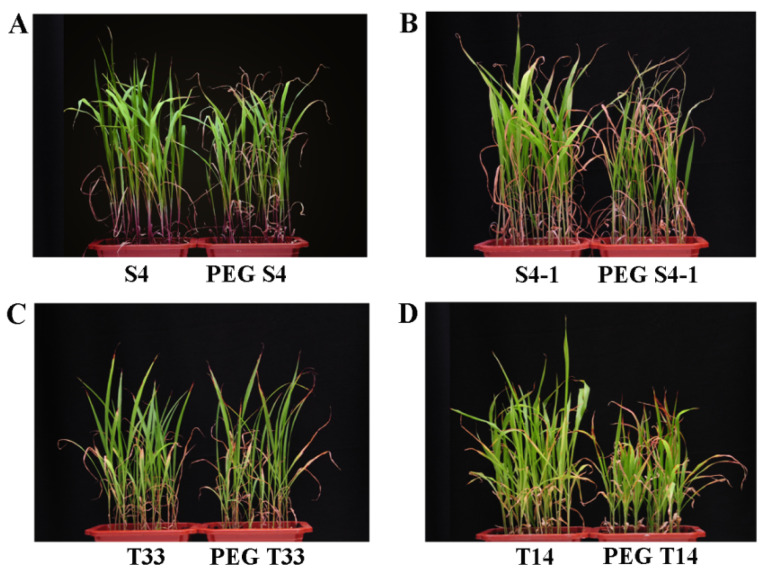
Phenotypic characteristics of the four sorghums varieties after 28 days of PEG treatment. Phenotypes of the sorghum S4/PEG S4 (**A**), S4-1/PEG S4-1 (**B**), T33/PEGT33 (**C**), and T14/PEG T14 (**D**) at the seedling stage. The left is the control group (S4, S4-1, T33, and T14), and the right is the PEG treatment group (PEG S4, PEG S4-1, PEGT33, and PEG T14).

**Figure 3 ijms-23-13297-f003:**
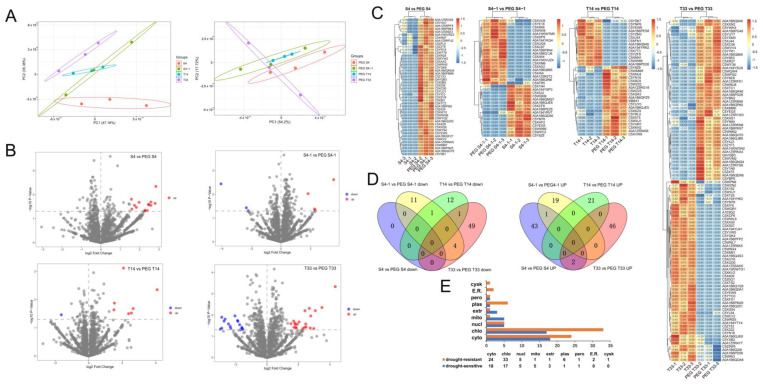
The bioinformatic analysis of DAPs in drought-sensitive (S4 and S4-1) and drought resistant (T14 and T33) sorghums. PCA plots from the full proteome of control and PEG-treated groups following PEG-simulated drought for 24 h (**A**). Volcano plots showing the DAPs of consensus proteins in S4, S4-1, T14, and T33 under drought stress (**B**). Heat maps displaying the log2 FC in abundance for DAPs significantly affected by drought stress. Each lane shows the mean of a biological replicate, and the analysis was performed in triplicate (**C**). Venn diagram showing the overlap in the numbers of the upregulated and downregulated proteins (**D**). Subcellular location analysis of DAPs of drought-sensitive and drought-resistant sorghums. Blue, yellow, green and red ovals represent up-or down-regulated proteins of S4 vs. PEG S4, S4-1 vs. PEG S4-1, T14 vs. PEG T14 and T33 vs. PEG T33, respectively. (**E**). cyto: cytoplasm; chlo: chloroplast; nucl: nucleus; mito: mitochondria; extr: extracellular; plas: plasma membrane; pero: peroxisome; E.R.: endoplasmic reticulum; cysk: cytoskeleton.

**Figure 4 ijms-23-13297-f004:**
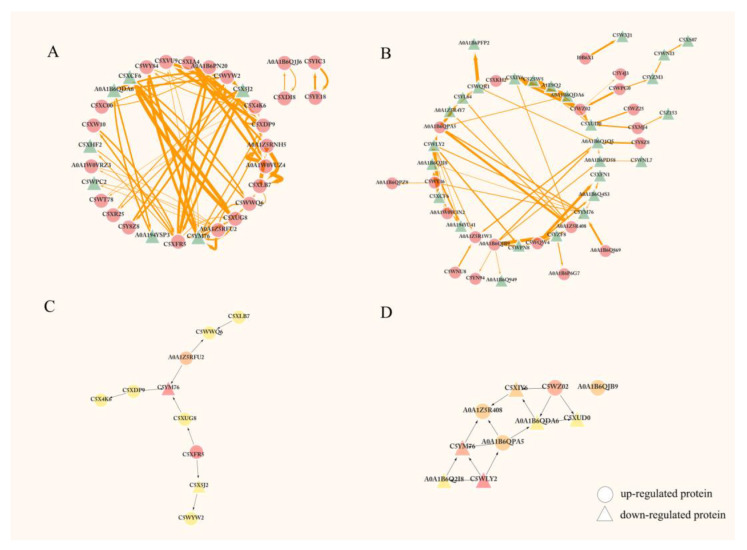
PPI network of 31 and 47 DAPs in drought-sensitive and drought-resistant sorghums, respectively. Functional correlation network of 31 DAPs of drought-sensitive sorghums (**A**). Functional correlation network of 47 DAPs of drought-resistant sorghums (**B**). The thickness of the connecting line between nodes represents the combined score value. The thicker it is, the stronger the interaction between the two proteins is. Green triangles and red circles represent down- and upregulated proteins in (**A**) and (**B**), respectively. The (**C**,**D**) represent the top ten DAP nodes in the networks of drought-sensitive sorghum and drought-resistant sorghums, respectively. The node color ranges from yellow to red, and the darker the node color, the more important it is in the interaction network. Triangles represent down-regulated proteins and circles represent up-regulated proteins in (**C**,**D**).

**Figure 5 ijms-23-13297-f005:**
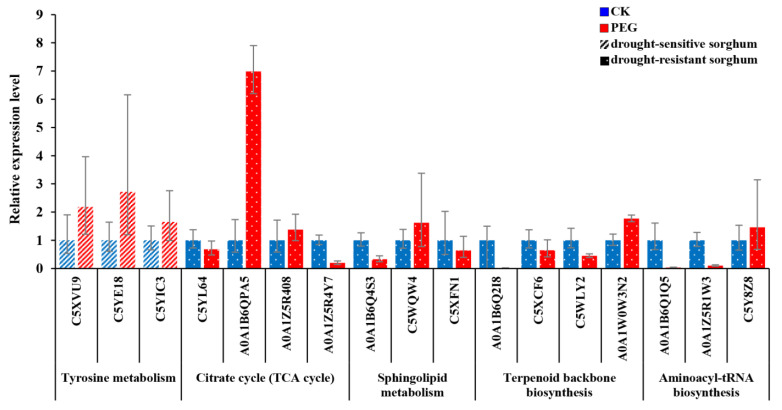
qRT-PCR validation of the mRNA expression profiles of 17 DAPs in drought-sensitive and drought-resistant varieties under control and PEG-stressed conditions. The average expression level in the control is set to 1. Error bars represent standard error (n = 3). C5XVU9: S-(hydroxymethyl)glutathione dehydrogenase; C5YE18: polyphenol oxidase (PPO); C5YIC3: PPO; C5YL64: pyruvate dehydrogenase E2 component; A0A1B6QPA5: phosphoenolpyruvate carboxykinase (PEPCK) (ATP); A0A1Z5R408: isocitrate dehydrogenase (IDH); A0A1Z5R4Y7: pyruvate dehydrogenase E2 component; A0A1B6Q4S3: neutral ceramidase (nCDase); C5WQW4: serine palmitoyltransferase (SPT); C5XFN1: sphinganine-1-phosphate (S1P) aldolase; A0A1B6Q2I8: 1-deoxy-D-xylulose-5-phosphate reductoisomerase; C5XCF6: geranylgeranyl diphosphate synthase (GGPS); C5WLY2: 4-hydroxy-3-methylbut-2-enyl diphosphate reductase (HDR); A0A1W0W3N2: polycis-polyprenyl diphosphate synthase (DHDDS); A0A1B6Q1Q5: aspartyl-tRNA synthetase; A0A1Z5R1W3: Histidyl-tRNA synthetase (HARS); C5Y8Z8: aspartyl-tRNA synthetase.

**Figure 6 ijms-23-13297-f006:**
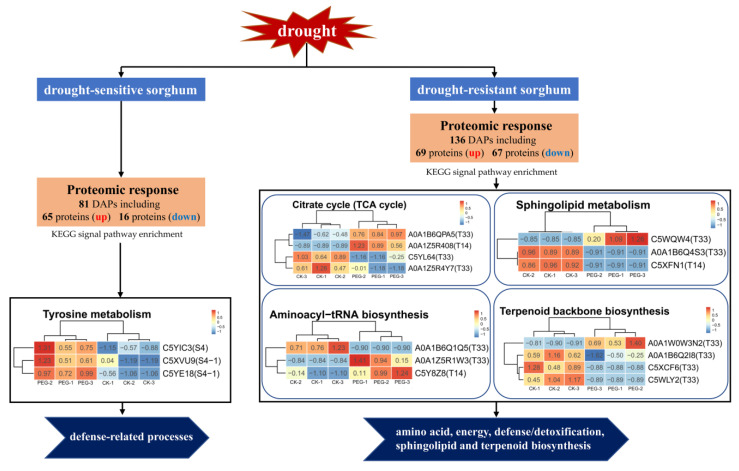
Model showing the proteomic responses of drought-sensitive and drought-resistant sorghum under drought stress.

**Table 1 ijms-23-13297-t001:** Significant enrichment analysis of DAP pathways (*p*-value < 0.05).

Pathway	*p*-Value	DAPs Name	KO	Regulated	Samples
Tyrosine metabolism	0.003564	C5XVU9	K00121	up	drought-sensitive
		C5YE18	K00422	up	
		C5YIC3	K00422	up	
Citrate cycle (TCA cycle)	0.000849	A0A1Z5R408	K00031	up	drought-resistant
		A0A1B6QPA5	K01610	up	
		C5YL64	K00627	down	
		A0A1Z5R4Y7	K00627	down	
Sphingolipid metabolism	0.004947	C5WQW4	K00654	up	
		A0A1B6Q4S3	K12349	down	
		C5XFN1	K01634	down	
Terpenoid backbone biosynthesis	0.001457	A0A1W0W3N2	K11778	up	
		C5XCF6	K13789	down	
		C5WLY2	K03527	down	
		A0A1B6Q2I8	K00099	down	
Aminoacyl-tRNA biosynthesis	0.012123	C5Y8Z8	K02433	up	
		A0A1Z5R1W3	K01892	up	
		A0A1B6Q1Q5	K01876	down	

## Data Availability

All the MS proteomics data have been deposited to the ProteomeXchange Consortium (http://proteomecentral.proteomexchange.org/cgi/GetDataset?ID=PXD033820, 1 September 2022) via the iProX partner repository [88] with the dataset identifier PXD033820.

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
