# Peer review of "Proteomic Analysis Revealed Different Molecular Mechanisms of Response to PEG Stress in Drought-Sensitive and Drought-Resistant Sorghums"

_ijms, 2022, doi:10.3390/ijms232113297_

Round 1
Reviewer 1 Report (Previous Reviewer 2)
The authors have addresses all my queries.
Author Response
Point 1: The authors have addresses all my queries.
Response: Thank you very much for your recognition and encouragement of our work.
Reviewer 2 Report (New Reviewer)
In this work, seedlings at the three-leaf stage of 4 cultivars (S4, S4-1, T33 and T14) of Sorghum bicolor were subjected to drought stress using 25% PEG-6000 treatment for 24 h and leaf proteins of untreated and treated seedlings were identified by mass spectrometry. Differentially abundant proteins (DAPs) were identified and compared between control ´ treatment within each cultivar as well as among different cultivars. The biochemical pathways in which the DAPs were down- or up-regulated were identified. Some genes (17) encoding important candidate proteins related to drought stress were selected and their expression profile was verified by quantitative PCR (qRT-PCR). The results reported in this work bring important data on the biochemical response of sorghum cultivars to drought stress. The study is suitable to be published in IJMS, but some relevant questions should be properly addressed before any further analysis.
Major points
1. Some important background information is missing about sorghum crop and drought stress. In which parts of the world this crop is cultivated? What are the main producers (provide the yield averages) of sorghum worldwide? Which parts of the plant are used and to what end? What is the impact of drought stress on sorghum yields? Which sorghum producing areas of the world are more affected by this abiotic stress?
2. Other previous works have investigated the proteomic response of sorghum to drought stress. Please make a clear and objective statement explaining how this work differs from the previous ones and what is new in comparison to other studies.
3. The proteomic response of sorghum to drought stress was investigated in drought-sensitive (S4 and S4-1) and drought-resistant (T33 and T14) cultivars. However, essential background information about these 4 cultivars is missing. What is the source of these cultivars? In which research institute they were bred? How were they classified as drought-sensitive or drought-resistant? Which physiological parameters were used to evaluate and classify the cultivars as drought-sensitive or drought-resistant? Is there any previous publication describing this evaluation?
4. At the beginning of the section 3.2, it would be very helpful to provide a clear and objective statement about which pathways or biological processes are enriched in the proteomic data sets.
5. Some sentences need to be carefully reevaluated. For example, see the sentence “Whatmore, the up-regulated S-(hydroxymethyl) glutathione (HMGSH) dehydrogenase protein (C5XVU9) was also enriched in tyrosine metabolism pathway under drought stress” (lines 422-424). If I understood correctly, the tyrosine metabolism pathway was enriched under drought stress, the S-(hydroxymethyl) glutathione (HMGSH) dehydrogenase protein (C5XVU9) being significantly upregulated in drought-sensitive cultivars.
6. The language of the manuscript could be further polished by a native English-speaker expert.
Minor points
1. The abbreviation of hour is h. I suggest using “24 h” instead of “24 hours”, for example. Please be consistent throughout the text.
2. The abbreviation of second or seconds is s (see the Guide for the Use of the International System of Units; physics.nist.gov/cuu/pdf/sp811.pdf). Please make the proper amendments in the text.
3. Lines 76-77. “Current proteomic studies of sorghum response to drought stress are extremely limited’. What does this statement really mean?
4. There are some other studies about the proteomic response of Sorghum bicolor under drought stress that were not cited: i) Proteomic analysis and interactions network in leaves of mycorrhizal and nonmycorrhizal sorghum plants under water deficit. Olalde-Portugal V, Cabrera-Ponce JL, Gastelum-Arellanez A, Guerrero-Rangel A, Winkler R, Valdés-Rodríguez S. PeerJ. 2020 Apr 23;8:e8991. doi: 10.7717/peerj.8991. eCollection 2020; ii) Identification of drought responsive proteins using gene ontology hierarchy. Sharma V, Sekhwal MK, Swami AK, Sarin R. Bioinformation. 2012;8(13):595-9. doi: 10.6026/97320630008595. Epub 2012 Jul 6.
5. Line 110. The word “content” is repeated twice: “relative water content (RWC) content”.
6. In Fig. 2, if I understood correctly, plants shown of the left side of each panel were not subjected to drought stress (controls). It would be helpful to make this distinction clear to the reader.
7. Line 370. Verify the word “creased”. Is this correct?
8. Line 690. Change “13,000 g” to “13,000 g” (italicize the letter g for centrifugal force, to differentiate it from g of grams).
9. Lines 693-695. I suggest the following text: Tryptic peptides were identified by nano-liquid chromatography (nLC) and tandem mass spectrometry (MS/MS) using an EASY-nLC 1000 system coupled to a Q Exactive Plus Orbitrap mass spectrometer (Thermo Fisher Scientific).
Author Response
Point 1: Some important background information is missing about sorghum crop and drought stress. In which parts of the world this crop is cultivated? What are the main producers (provide the yield averages) of sorghum worldwide? Which parts of the plant are used and to what end? What is the impact of drought stress on sorghum yields? Which sorghum producing areas of the world are more affected by this abiotic stress?
Response 1: Thanks for this good suggestion. According to this suggestion, we supplemented and improved the relevant content, which is as follows: “Sorghum (Sorghum bicolor (L.) Moench) is the fifth most important grain crop in the world and is mainly distributed in arid and semi-arid tropical regions of the world [1]. Global sorghum production in 2021 was 62.167 million tons, with Africa being the largest producing region, accounting for 40% of the world's total, followed by the United States (https://www.usda.gov/). Sorghum has a wide range of uses, such as eating, feeding, brewing, bioenergy, and chemical materials [2]. Severe drought stress will lead to shorter seedling stage, smaller ear head, poor fruit maturity, greatly reduced yield or no harvest. Drought is one of the major constraints on sorghum production, especially in African and Asian countries [3]. Severe drought stress leads to a shorter seedling stage, a smaller ear head, poor fruit maturity, a greatly reduced yield, or no harvest. Thus, it is of great significance to study the effects of drought stress on the growth and physiology of sorghum plants and to improve the drought resistance and yield of sorghum.” (Line 45-56)
Point 2: Other previous works have investigated the proteomic response of sorghum to drought stress. Please make a clear and objective statement explaining how this work differs from the previous ones and what is new in comparison to other studies.
Response 2: Thanks for this good suggestion. In this study, we summarized and learned from previous research works and carried out some innovative studies on this basis [4]. The similarities and differences between this work and previous studies are shown in the following table 1. First, in the selection of test materials, we selected two drought-resistant and drought-sensitive sorghum cultivars, respectively. We not only analyzed the characteristics of different drought-response materials at the proteome level, but also sorted out the common drought-response pathways and proteins in drought-resistant and drought-sensitive materials. The results of this study showed that drought-resistant and drought-sensitive sorghums adopted different strategies in response to drought stress, which was consistent with the previous conclusions [4]. However, the specific strategies of these sorghums in response to drought stress were different, which provided a new clue for the study of drought resistance mechanism. In addition, in terms of research methods, we adopted nano LC-MS/MS method combined with PRM mass spectrometry verification to further improve the accuracy and effectiveness of proteomic data. Of course, if more samples could be extracted at different times during PEG stress treatment and combined with transcriptome data, it would be more helpful to conduct a deeper and detailed analysis of the drought resistance mechanism of sorghum.
Table 1. Summary of the similarities and differences between this work and previous proteomic studies of sorghum under drought stress [4].
|
S. bicolor Variety with known Drought Phenotype |
Plant Tissues |
Drought Experiment |
Techniques Used |
Summary of Key Findings |
|
11434 — tolerant 11431—susceptible |
Leaves |
Withholding water from seedlings until soil water potential of 1 MPa, Re-watering for 24 h. |
2D-DIGE, MALDI-TOF-MS |
Transcription, protein synthesis, protein destination and storage—related proteins were generally more up-regulated in the drought-tolerant varieties than the sensitive type in response to drought and/or re-watering. Proteases were up-regulated in the drought-sensitive variety in response to water deprivation. |
|
SA1441— tolerant ICSB338—susceptible |
Roots |
Withholding water from seedlings for 8 days. |
iTRAQ, qRT-PCR |
Common and unique drought-responsive proteins were identified in the two varieties. The tolerant SA1441 up-regulated transcription, protein synthesis, protease inhibitors, signaling transduction, and transporter-related proteins in response to water deprivation. The sensitive ICSB338 down-regulated metabolism and protein synthesis but increased the proteolysis. |
|
BTx623 |
Roots |
20% PEG-6000 applied on 16-day-old seedlings growing on nutrient medium over 24 h. |
2D-PAGE, CBB-G250 staining, MALDI-TOF-TOF MS |
Up-regulated proteins were mainly involved in carbohydrate /energy /lipid metabolism, antioxidant functions, stress response (LEA like-proteins), protein synthesis and transport, regulation of transcription, and signaling functions. |
|
EI9 — tolerant Tabat — sensitive |
Leaves |
Withholding water from 14-day old seedlings for 7 days. |
Nanoflow UPLC, MS |
36 proteins were detected. Of these, 23 were drought-induced in either one or both sorghum varieties. Identified proteins were involved in a range of functions, including response to stress, metabolic processes, photosynthesis, cell wall biosynthesis/degradation, and fatty acid biosynthesis. |
|
S4 and S4-1 —drought-sensitive; T33 and T14 —drought-resistant |
Leaves |
25% PEG-6000 treatment on seedlings at the three-leaf stage for 24 h. |
Nano LC-MS/MS, PRM, qRT-PCR |
The response mechanism of the drought-sensitive sorghum leaves to drought was attributed to the up-regulation of proteins involved in the tyrosine metabolism pathway with defense functions. Drought-resistant sorghum leaves respond to drought by promoting the TCA cycle, enhancing sphingolipid biosynthesis, interfering with triterpenoid metabolite synthesis, and influencing of aminoacyl-tRNA biosynthesis. The 17 important candidate proteins related to drought stress have been identified in this study. |
Point 3: The proteomic response of sorghum to drought stress was investigated in drought-sensitive (S4 and S4-1) and drought-resistant (T33 and T14) cultivars. However, essential background information about these 4 cultivars is missing. What is the source of these cultivars? In which research institute they were bred? How were they classified as drought-sensitive or drought-resistant? Which physiological parameters were used to evaluate and classify the cultivars as drought-sensitive or drought-resistant? Is there any previous publication describing this evaluation?
Response 3: Thanks for this suggestion. The supplement and modifications of these 4 varieties are as follows: “The drought-sensitive (S4 (Heibeinuoza 5) and S4-1(Qianjinchui)) and drought-resistant (T33 (Theis) and T14 (Rio)) sorghum varieties were obtained from the Germplasm Bank of the Institute of Crop Sciences, Chinese Academy of Agricultural Sciences. Additionally, their ability to respond to drought stress has been identified by multilevel phenotypic analysis. The screening indicators and identification methods of drought resistance of sorghums were described in our previous work [5].” (Line 677-682)
Point 4: At the beginning of the section 3.2, it would be very helpful to provide a clear and objective statement about which pathways or biological processes are enriched in the proteomic data sets.
Response 4: Thanks for this suggestion. The modifications at the beginning of the section 3.2 are as follows: “In this study, the response of drought-sensitive sorghum leaves to PEG stress was mainly enriched in the tyrosine metabolism pathway, among which the three proteins C5XVU9, C5YE18, and C5YIC3 were significantly upregulated, which played an important role in the drought stress response.” (Line 416-419)
Point 5: Some sentences need to be carefully reevaluated. For example, see the sentence “Whatmore, the up-regulated S-(hydroxymethyl) glutathione (HMGSH) dehydrogenase protein (C5XVU9) was also enriched in tyrosine metabolism pathway under drought stress” (lines 422-424). If I understood correctly, the tyrosine metabolism pathway was enriched under drought stress, the S-(hydroxymethyl) glutathione (HMGSH) dehydrogenase protein (C5XVU9) being significantly upregulated in drought-sensitive cultivars.
Response 5: Thanks for this suggestion. This sentence has been corrected as follows: “Furthermore, the tyrosine metabolism pathway was enriched under drought stress, in which the S-(hydroxymethyl) glutathione (HMGSH) dehydrogenase protein (C5XVU9) was significantly upregulated in drought-sensitive varieties.” (Lines 438-440).
Point 6: The language of the manuscript could be further polished by a native English-speaker expert.
Response 6: Thanks for this suggestion. The language of this manuscript has been polished by an experienced, native English-speaking editor recommended by MDPI.
Minor points
Point 1: The abbreviation of hour is h. I suggest using “24 h” instead of “24 hours”, for example. Please be consistent throughout the text.
Response 1: Thanks for this suggestion. The word “hours” has been changed into “h” in full text.
Point 2: The abbreviation of second or seconds is s (see the Guide for the Use of the International System of Units; physics.nist.gov/cuu/pdf/sp811.pdf). Please make the proper amendments in the text.
Response 2: Thanks for this suggestion. The word “sec” has been corrected to “s” in full text.
Point 3: Lines 76-77. “Current proteomic studies of sorghum response to drought stress are extremely limited’. What does this statement really mean?
Response 3: Thanks for this suggestion. This sentence has been corrected to “At present, the proteomic studies of sorghum response to drought stress mainly includes the following aspects.” (Line 82-83)
Point 4: There are some other studies about the proteomic response of Sorghum bicolor under drought stress that were not cited: i) Proteomic analysis and interactions network in leaves of mycorrhizal and nonmycorrhizal sorghum plants under water deficit. Olalde-Portugal V, Cabrera-Ponce JL, Gastelum-Arellanez A, Guerrero-Rangel A, Winkler R, Valdés-Rodríguez S. PeerJ. 2020 Apr 23;8:e8991. doi: 10.7717/peerj.8991. eCollection 2020; ii) Identification of drought responsive proteins using gene ontology hierarchy. Sharma V, Sekhwal MK, Swami AK, Sarin R. Bioinformation. 2012;8(13):595-9. doi: 10.6026/97320630008595. Epub 2012 Jul 6. 4.
Response 4: Thanks for this suggestion. The article (i Proteomic analysis and interactions network in leaves of mycorrhizal and nonmycorrhizal sorghum plants under water deficit. Olalde-Portugal V, Cabrera-Ponce JL, Gastelum-Arellanez A, Guerrero-Rangel A, Winkler R, Valdés-Rodríguez S. Peer J. 2020 Apr 23;8: e8991. doi: 10.7717/peerj.8991.) has been quoted on the section of introduction as follows: “Furthermore, proteomic analyses indicate that mycorrhizal and nonmycorrhizal sorghum plants use different molecular mechanisms to deal with water deficit stress [6]” (Line 105-107). The article (ii Identification of drought responsive proteins using gene ontology hierarchy. Sharma V, Sekhwal MK, Swami AK, Sarin R. Bioinformation. 2012;8(13):595-9. doi: 10.6026/97320630008595.) has been quoted on the section of Materials and Methods (line 744).
Point 5: Line 110. The word “content” is repeated twice: “relative water content (RWC) content”.
Response 5: Thanks for this suggestion. The word “relative water content (RWC) content” has been corrected to “relative water content (RWC)”. (Line 43)
Point 6: In Fig. 2, if I understood correctly, plants shown of the left side of each panel were not subjected to drought stress (controls). It would be helpful to make this distinction clear to the reader.
Response 6: Thanks for this suggestion. In Fig. 2, we have added the description as follows: “The left is the control group (S4, S4-1, T33, and T14), and the right is the PEG treatment group (PEG S4, PEG S4-1, PEGT33, and PEG T14).” (Line 176-177)
Point 7: Line 370. Verify the word “creased”. Is this correct?
Response 7: Thanks for this suggestion. The word “creased” has been corrected to “increased”. (Line 382)
Point 8: Line 690. Change “13,000 g” to “13,000 g” (italicize the letter g for centrifugal force, to differentiate it from g of grams).
Response 8: Thanks for this suggestion. The word “13,000 g” has been corrected to “13,000 g.” (Line 714)
Point 9: Lines 693-695. I suggest the following text: Tryptic peptides were identified by nano-liquid chromatography (nLC) and tandem mass spectrometry (MS/MS) using an EASY-nLC 1000 system coupled to a Q Exactive Plus Orbitrap mass spectrometer (Thermo Fisher Scientific).
Response 9: Thanks for this suggestion. This sentence “The isolation and Nano LC-MS/MS analysis of peptide fragments was performed by using liquid chromatography system Easy nLC 1000 coupled with the MS system Q-Excative Plus (Thermo Fisher Scientific).” has been corrected to “Tryptic peptides were identified by nano-liquid chromatography (nLC) and tandem mass spectrometry (MS/MS) using an EASY-nLC 1000 system coupled to a Q Exactive Plus Orbitrap mass spectrometer (Thermo Fisher Scientific).” (Line 716-718)
References:
- Ananda, G. K. S.; Myrans, H.; Norton, S. L.; Gleadow, R.; Furtado, A.; Henry, R. J., Wild Sorghum as a Promising Resource for Crop Improvement. Frontiers in plant science 2020, 11, 1108.
- Abreha, K. B.; Enyew, M.; Carlsson, A. S.; Vetukuri, R. R.; Feyissa, T.; Motlhaodi, T.; Ng'uni, D.; Geleta, M., Sorghum in dryland: morphological, physiological, and molecular responses of sorghum under drought stress. Planta 2021, 255, (1), 20.
- Massey, A. R. Sweet sorghum (Sorghum bicolor) biomass, generated from biofuel production, as a reservoir of bioactive compounds for human health. Colorado State University., 2014.
- Ngara, R.; Goche, T.; Swanevelder, D. Z. H.; Chivasa, S., Sorghum's Whole-Plant Transcriptome and Proteome Responses to Drought Stress: A Review. Life (Basel, Switzerland) 2021, 11, (7).
- Zhang, X. X. P., Y. H.; Ren, F. L.; Pu, W. J.; Wang, D. P.; Li, Y. B.; Lu, P.; Li, G. Y.; Zhu, L., Establishment of an accurate evaluation method for drought resistance based on multilevel phenotype analysis in sorghum. Acta Agronomica Sinica 2019, 45, (11), 1735-1745.
- Olalde-Portugal, V.; Cabrera-Ponce, J. L.; Gastelum-Arellanez, A.; Guerrero-Rangel, A.; Winkler, R.; Valdés-Rodríguez, S., Proteomic analysis and interactions network in leaves of mycorrhizal and nonmycorrhizal sorghum plants under water deficit. PeerJ 2020, 8, e8991.
Reviewer 3 Report (New Reviewer)
The work deals with using a proteomic approach to study molecular mechanisms of PEG stress in sorghum samples.
The changes of protein expression profiles during drought-stress were compared in drought-sensitive and drought-resistant sorghums.
The 17 important protein candidates revealed to be related to drought stress and validated by PCR approach.
REMARKS:
1 In introduction, 3rd paragraph, add an important citation published recently applying proteomic analysis.
“Proteomics is an emerging subject and hotspot in functional genomics research in the post-genomic era. It can clarify the biological functions of proteins expressed in the genome that perform life activities [7, https://doi.org/10.1515/biol-2019-0070].“
2 In materials and methods, draw a scheme of experimental workflow and provide it as a picture to show a reader principle of proteomic sample preparation in a quick way
3 In conclusion, provide a shortly future aims of the authors in this area of investigation. What else could be done in the proteomic investigation of drought stress related factors in plant samples.
Author Response
Point 1: In introduction, 3rd paragraph, add an important citation published recently applying proteomic analysis.
“Proteomics is an emerging subject and hotspot in functional genomics research in the post-genomic era. It can clarify the biological functions of proteins expressed in the genome that perform life activities [7, https://doi.org/10.1515/biol-2019-0070].”
Response 1: Thanks for this suggestion. The important citation has been added in line 78.
Point 2: In materials and methods, draw a scheme of experimental workflow and provide it as a picture to show a reader principle of proteomic sample preparation in a quick way.
Response 2: Thanks for this suggestion. The flowchart of the proteomics analysis of drought-sensitive (S4 and S4-1) and drought-resistant (T33 and T14) sorghum varieties has been supplemented in the section of Materials and Methods shown as Figure S2. (Line 772-773)
Fig S2. The flowchart of proteomics analysis of drought-sensitive (S4 and S4-1) and drought-resistant (T33 and T14) sorghum varieties in this study.
Point 3: In conclusion, provide a shortly future aims of the authors in this area of investigation. What else could be done in the proteomic investigation of drought stress related factors in plant samples.
Response 3: Thanks for this good suggestion. Further studies will focus on verifying the specific biological functions and mechanisms of these 17 key candidate proteins in response to drought stress in sorghum so as to provide genetic resources for the molecular breeding of sorghum drought resistance. In the future, proteomics can be combined with post-translational modification omics, transcriptomics, and metabolomics to deeply explore species characteristics and signal regulatory networks. (Line 794-799)

Round 2
Reviewer 3 Report (New Reviewer)
Authors have reacted to the given queries.
This manuscript is a resubmission of an earlier submission. The following is a list of the peer review reports and author responses from that submission.
Round 1
Reviewer 1 Report
The authors have done a good job in this study, although, its a well-executed study, I think a few things in the manuscript should be improved, below my comments.
Figure 1 Bar colors do not match, please correct this.
Figure2: Please change the legends, chlorophyll content is not shown in those figures.
Can authors try to measure the plant's fresh weight?
Figure 4: I would change the presentation of figure 4, please refer to Mcloughlin et al. 2020;2021 (Nature plants)
Can authors represent their data in volcano plot? Thereby highlighting the differential up or down-regulated protein candidates
As the authors mention that their data is in triplicates, I think the representation through a volcano or other plots that can depict the p-Value should be implemented.
Can authors try to use (if) available transcript data for drought-sensitive and tolerant sorghum to see how much of their transcript correlates with the proteome?
It is a good practice to show the PCA plot to the analyze the integrity of the data points,
I cannot fully follow the discussion, I think it does not justify well the figure 7 or the conclusion model.
It will be good to have the family/cluster of proteins that are differentially expressed in model 7, to be depicted via the heat map analyses
Reviewer 2 Report
The present manuscript titled "Proteomic analysis revealed different molecular mechanisms of response to PEG stress in drought-sensitive and resistant sorghums" investigated the function of protein-responsive proteins, clarified the molecular basis of the drought resistance differences in different drought-resistance sorghum, and explored key candidate genes and pathways related to drought resistance and breeding drought-tolerant cultivars.
The manuscript was well-written with valid materials and methods and results. Results were well explained.
I just feel that the experiment could be done with a few more high doses to PEG to see the effects of varying degree of drought stress on sorghum.
4.2. and 4.3. have the same heading. Needs to be changed.